# Potential traumatic events in the workplace and depression, anxiety and post-traumatic stress: a cross-sectional study among Dutch gynaecologists, paediatricians and orthopaedic surgeons

Karel Willem Frank Scheepstra ![ORCID],[1] Hannah S Pauw,[2] Minouk Esmee van Steijn,[3,4] Claire A I Stramrood,[3] Miranda Olff,[1,5] Maria G van Pampus ![ORCID] [3]

For numbered affiliations see end of article.

**Correspondence to**
Karel Willem Frank Scheepstra;
k.w.scheepstra@
amsterdamumc.nl

## ABSTRACT

**Objective** To compare the prevalence of work-related potential traumatic events (PTEs), support protocols and mental health symptoms across Dutch gynaecologists, orthopaedic surgeons and paediatricians.

**Design** Cross-sectional study, supplementary analysis of combined data.

**Setting** Nationwide survey between 2014 and 2017.

**Participants** An online questionnaire was sent to all Dutch gynaecologists, orthopaedic surgeons and paediatricians, including resident physicians (4959 physicians). 1374 questionnaires were eligible for analysis, corresponding with a response rate of 27.7%.

**Outcome measures** Primary outcome measures were the prevalence of work-related PTEs, depression, anxiety, psychological distress and traumatic stress, measured with validated screening instruments (Hospital Anxiety and Depression Scale, Trauma Screening Questionnaire). Secondary outcomes were the association of mental health and defensive practice to traumatic events and support protocols.

**Results** Of the respondents, 20.8% experienced a work-related PTE at least 4 weeks ago. Prevalence rates indicative of depression, anxiety or post-traumatic stress disorder (PTSD) were 6.4%, 13.6% and 1.5%, respectively. Depression (9.2% vs 5.2%, p=0.019), anxiety (18.2% vs 8.2%, p<0.001) and psychological distress (22.8% vs 12.5%, p<0.001) were significantly more prevalent in female compared with male attendings. The absence of a support protocol was significantly associated with more probable PTSD (p=0.022). Those who witnessed a PTE, reported more defensive work changes (28.0% vs 20.5%, p=0.007) and those with probable PTSD considered to quit medical work more often (60.0% vs 35.8%, p=0.032).

**Conclusion** Physicians are frequently exposed to PTEs with high emotional impact over the course of their career. Lacking a support protocol after adverse events was associated with more post-traumatic stress. Adverse events were associated with considering to quit medical practice and a more defensive practice. More awareness must be created for the mental health of physicians as well as for the implementation of a well-organised support system after PTEs.

## Strengths and limitations of this study

► This is one of the few studies done on post-traumatic stress disorder (PTSD) in physicians, in particular with regard to mental health in relation to social and occupational functioning and clinical practice related consequences.
► This study incorporated the A and E criterion in our probable PTSD diagnosis, therefore, the prevalence is more likely to reflect the real prevalence of PTSD.
► One limitation is that the population of this study is limited to three specialties and one country in Europe, and thus, it is difficult to translate these results to physicians worldwide.

## INTRODUCTION

Physicians are frequently exposed to medical adverse events, such as life-threatening situations, illnesses, grief, death and patient violence and aggression. Dealing with such events makes the medical profession one with occupational health hazards, comparable with fire fighters, rescue workers, military and police personnel.[1–3] The support for physicians is, however, underdeveloped in most areas of the medical profession compared with other professions.[4 5] It is generally assumed that healthcare providers have adequate coping mechanisms to deal with such work stressors. However, adequate coping mechanisms do not warrant successful coping with severe potential traumatic events (PTEs). Our previous work showed that most gynaecologists, paediatricians and midwives experience support and protocols after work-related PTEs as insufficient.[6 7] Dutch gynaecologists reported that development of coping strategies with work-related stressors are neither taught during graduate nor in specialist training.[8] Physician health is one of

the competencies described in the CanMEDS framework used in Dutch medical education.[9] This framework recognises that, to provide optimal patient care, physicians must take responsibility for their own well-being and that of their colleagues. Several hospitals have started peer-support programmes, as this was shown to help physicians dealing with adverse events.[10]

### Depression, anxiety and post-traumatic stress disorder

Major depressive disorder (MDD) is a common affective disorder that is characterised by a depressed mood and/or loss of pleasure.[11] A review on the epidemiology of mental disorders in Europe described a 1-year prevalence of 6.9% in the general population[12]; the Nemesis-2 study showed a 1-year prevalence of mood disorders of 3.0% among Dutch citizens with a high income.[13] A meta-analysis on depressive symptoms and MDD among resident physicians showed high prevalence rates (20.9%–43.2%).[14] Fewer studies were done on depressive symptoms among medical specialists, showing prevalence rates from 10.3% to 28.0%.[15–18] A recent literature review showed that as many as 40 per 100 000 physicians die by suicide each year in the USA, more than double the rate of the general population.[19]

Anxiety disorders include many different disorders, ranging from a specific phobia to a generalised anxiety disorder. The 1-year prevalence for anxiety disorders in the Netherlands is 6.0% among citizens with a high income and 10.1% among citizens with a low income.[13]

Post-traumatic stress disorder (PTSD) is a highly debilitating disorder that can develop after exposure to a psychotraumatic event.[20] Symptoms of PTSD include intrusions, avoidance of stimuli related to the trauma, alterations in cognition or mood and hyperarousal. The point prevalence of post-traumatic stress (PTS) symptoms among physicians ranges between 3.8% and 15.0.[15 21–25]

The main objective of this supplementary analysis was to compare the prevalence of PTEs and related mental health disorders between a large group of gynaecologists, paediatricians and orthopaedic surgeons. Additionally, this study investigates PTEs and clinical practice-related consequences. Combining the data of previous work allows us to gain statistical power to explore the association between support protocols and mental health symptoms in a larger group.

## METHODS
### Population and procedure

Each physician received one invitation for an online questionnaire (online supplementary file 1) and two reminders over a period of 8 weeks. The questionnaire was sent through Thesistools or Surveymonkey, creating an anonymous (non-traceable) email link. The questionnaire was sent to 1589 gynaecologists, 2160 paediatricians and 1210 orthopaedic surgeons, with a total of 4959 physicians, including residents. The invitation to participate in the questionnaire for the gynaecologists was sent

in 2014, orthopaedic surgeons in 2016 and paediatricians in 2017. This is an additional analysis of combining data of previous work.

The membership databases of the Dutch Society of Obstetrics & Gynaecology, Dutch Society of Orthopaedic Surgeons and the Dutch Society of Paediatricians were used for the invitation. Registration as a medical specialist or resident in one of these societies is obligatory and therefore most Dutch medical specialists in these fields were contacted. These databases contained residents, attending, retired and non-practising physicians.

### Patient and public involvement

Engagement with the described medical societies was vital to ensure participation and a high response rate. The qualitative questions were piloted among several medical specialists of each specific medical specialty.

### Measurements

The instruments included the Hospital Anxiety and Depression Scale (HADS) and the Trauma Screening Questionnaire (TSQ) as validated screening instruments. The HADS is a screening instrument for depression and anxiety with 14 items (4-point Likert scale), where the subscales contain seven questions each. The cut-off value of the Dutch version of the depression (HADS-D) and anxiety subscale (HADS-A) is equal to or higher than eight. The total HADS cut-off value for general distress is equal or higher than 12, corresponding to clinically relevant psychological distress; this is a rather non-specific but sensitive measurement to screen for mental disorders.[26] Cut-off scores were used according to these validation studies. The sensitivity and specificity for both HADS-A and HADS-D are approximately 0.80, with Cronbach's alphas for HADS-A from 0.68 to 0.93 (mean 0.83) and for HADS-D from 0.67 to 0.90, in different populations.[27]

The TSQ is a 10-item screening instrument for a probable diagnosis of PTSD according to the DSM-IV, with a sensitivity of 0.94 and specificity of 0.56 when using a cut-off of equal to 6 or higher. The Dutch psychometric properties are validated with Cronbach's alphas from 0.71 to 0.91.[28] Only respondents who answered 'yes' to experiencing a work-related traumatic event at least 4 weeks ago were asked to fill out the TSQ.

Additionally, the survey contained questions about demographics, personal experiences, coping and support strategies after adverse events in the workplace. Regarding the latter, multiple-choice options were given as well as an open field where respondents were able to add their individual answers or comment on their experience. These qualitative questions were piloted among several medical specialists of all three medical specialties, to increase face value of the questions. To quantify the number of PTEs and previous PTS symptoms, questions were added based on the criteria of the diagnosis of PTSD in the DSM-IV (questions 22 and 35, online supplementary file 1).

**Table 1**  Respondent characteristics

|  | Total (n=4959) | Gynaecologists (n=1589) | Orthopaedic surgeons (n=1210) | Paediatricians (n=2160) |
|---|---|---|---|---|
| Response rate | 27.7 | 48.9 | 21.3 | 29.8 |
| Gender |  |  |  |  |
| Male | 44.9 | 34.7 | 85.6 | 32.7 |
| Female | 55.1 | 65.3 | 14.4 | 67.3 |
| Position |  |  |  |  |
| Resident | 23 | 26.9 | 20.9 | 18 |
| Attending physician | 67.2 | 64.7 | 62.3 | 74.9 |
| Retired | 7.1 | 5.4 | 13.4 | 5.6 |
| Non-practising | 2.6 | 3 | 3.4 | 1.5 |
| Years in practice |  |  |  |  |
| ≤5 | 12.4 | 14.3 | 12 | 9.8 |
| 10-June | 18.6 | 19 | 19.5 | 17.3 |
| 15-November | 15.8 | 17.7 | 14 | 13.9 |
| 16–20 | 14.3 | 11.6 | 13.4 | 19.3 |
| 21–25 | 12.4 | 12.8 | 9.2 | 13.9 |
| 26–30 | 11.3 | 9.5 | 14.4 | 12 |
| >30 | 15.1 | 14.7 | 17.5 | 13.9 |
|  |  | * |  |  |
| Complaints disciplinary board | 19.9 | 21 | 29.8 | 11.2 |
| Support protocol |  |  |  |  |
| Yes | 19.6 | 15.8 | 18.8 | 26.3 |
| No | 42.4 | 52.7 | 30.1 | 34.1 |
| I do not know | 38 | 31.4 | 51 | 39.5 |

All values shown as %.

*2 missing.

## Statistical analysis

Statistical analysis was performed using IBM Statistical Package for the Social Sciences (SPSS) V.22 and JASP V.0.8.5.1. Demographic data and multiple-choice questions were analysed using descriptive statistics and exported as frequency tables and bar charts. Differences in outcomes (in various groups) for categorical variables were tested using either a $X^2$ test or a Fisher's exact test where applicable. A two-sided p≤0.05 was considered statistically significant. A multivariable logistic regression analysis was performed to estimate the effects of age, work experience, gender and position on the likelihood that participants develop mental health outcomes, a p≤0.05 was considered statistically significant. Multiple testing corrections were not performed.

## RESULTS

A total of 1374 questionnaires were collected with a response rate of 27.7%. Table 1 shows the respondents characteristics by specialty.

The majority of the orthopaedic surgeons were male (85.6%) while in the other specialties the majority were female. In all specialties, most of the respondents (62.3%–74.9%) were attending physicians, as seen in table 1. The gender and position distributions were compared with the membership databases of the societies, and for none of the specialties the distribution in the sample was significantly different from the population. The orthopaedic surgeons reported to have had the most complaints at a disciplinary board (29.8%). One hundred and eight (26.3%) paediatricians reported to have a support protocol, whereas only 18.8% of the orthopaedic surgeons and 15.8% of the gynaecologists reported to have a protocol after adverse events.

Outcomes of the HADS and TSQ by specialty are shown in table 2.

Eighty-eight (6.4%) respondents scored above the cut-off value for depression and one hundred and eighty eight (13.6%) respondents scored above the cut-off value for anxiety. Two hundred and thirty-six (17.1%) scored above the cut-off value of the total HADS, corresponding to clinically relevant psychological distress. When comparing specialties, there is no significant difference in depression. However, looking at the prevalence

**Table 2** Depression, anxiety, psychological distress and PTSD by specialty

| | Total | Gynaecologists | Orthopaedic surgeons | Paediatricians |
|---|---|---|---|---|
| | (n=1374) | (n=672) | (n=292) | (n=410) |
| Depression | 6.4 | 6.5 | 4.8 | 7.3 |
| Anxiety | 13.6 | 15.8 | 8.2 | 14.1 |
| Psychological distress | 17.1 | 18.2 | 12 | 19.3 |
| PTSD | | | | |
| PTE ≥4 weeks ago | 20.8 | 12.8 | 19.5 | 34.9 |
| PTS symptoms earlier | 13.4 | 8.2 | 8.6 | 25.4 |
| TSQ | 1.5 | 1.5 | 0.3 | 2.2 |

All values shown in % scored above HADS or TSQ cut-off.
HADS, Hospital Anxiety and Depression Scale; PTE, potential traumatic event; PTS, post-traumatic stress; PTSD, post-traumatic stress disorder; TSQ, Trauma Screening Questionnaire.

of anxiety, the orthopaedic surgeons scored significantly lower compared with both gynaecologists (15.8% vs 8.2%, $\chi^2$, p<0.01) and paediatricians (14.1% vs 8.2%, $\chi^2$, p<0.05). Also psychological distress was significantly lower compared with gynaecologists (18.2% vs 12.0%, $\chi^2$, p<0.05) and paediatricians (19.3% vs 12.0%, $\chi^2$, p<0.01).

As seen in table 3, no significant difference was found between male and female residents in rates of depression, anxiety and psychological distress. When comparing gender among attendings, the rates of depression (5.2% vs 9.2%, p=0.019), anxiety (8.2% vs 18.2%, p<0.001) and the total HADS (12.5.0% vs 22.8%, p<0.001) were significantly higher among female attendings. When comparing among all respondents, the prevalence rates of anxiety (10.1% vs 18.7%, p<0.001) and psychological distress (14.2% vs 22.3%, p<0.001) were significantly higher among female physicians There were no differences in

primary outcomes when comparing residents with attendings. Turning to the results of the logistic regression analysis, there was a significant increase in the prevalence of depression in the groups with more years in practice (OR 2.95, p=0.030, 95% CI 1.11 to 7.80, online supplementary file 2), but not with increased age. We found a significant decrease in the prevalence of anxiety with more years in practice (OR 0.33, p=0.001, 95% CI 0.17 to 0.64, online supplementary file 2). The highest rate of respondents with a score above the cut-off value for anxiety were found among physicians with 6–10 years in practice (online supplementary file 2).

### Post-traumatic stress disorder
Two hundred and eighty-six respondents (20.8%) reported having experienced at least one PTE during their work, at least 4 weeks ago, thereby meeting the Diagnostic and

**Table 3** Depression, anxiety and psychological distress by position (resident vs attending) and gender

| | Total (n=1240) | Depression | P value | Anxiety | P value | Psychological distress | P value |
|---|---|---|---|---|---|---|---|
| Gender | | | | | | | |
| Male | 40.8 | 5.5 | 0.075 | 10.1 | **<0.001** | 14.2 | **<0.001** |
| Female | 59.2 | 8.2 | | 18.7 | | 22.3 | |
| Position | | | | | | | |
| Resident | 25.4 | 4.7 | 0.112 | 17.1 | 0.125 | 18.4 | 0.887 |
| Attending | 74.6 | 7.4 | | 13.6 | | 18.1 | |
| Residents | | | | | | | |
| Male | 25.6 | 3.7 | 0.346 | 13.6 | 0.053 | 14.8 | 0.058 |
| Female | 74.4 | 5.1 | | 18.3 | | 19.6 | |
| Attending | | | | | | | |
| Male | 45.9 | 5.2 | **0.008** | 8.2 | **<0.001** | 12.5 | **<0.001** |
| Female | 54.1 | 9.2 | | 18.2 | | 22.8 | |

All values shown as % scored above cut-off HADS.
All differences were analysed using $\chi^2$.
Significant differences in bold.
HADS, Hospital Anxiety and Depression Scale.

**Table 4** Post-traumatic stress outcomes by position (resident vs attending) and gender

| | Total (n=1240) | PTE | P value | PTS symptoms Earlier | P value | PTSD | P value |
|---|---|---|---|---|---|---|---|
| **Gender** | | | | | | | |
| Male | 40.8 | 21.3 | 0.878 | 11.3 | 0.044 | 1.2 | 0.516 |
| Female | 59.2 | 21 | | 15.3 | | 1.6 | |
| **Position** | | | | | | | |
| Resident | 25.4 | 16.8 | 0.03 | 7.9 | <0.001 | 0.9 | 0.391 |
| Attending | 74.6 | 22.6 | | 15.6 | | 1.6 | |
| **Residents** | | | | | | | |
| Male | 25.6 | 21 | 0.245 | 4.9 | 0.246 | 0 | 0.306 |
| Female | 74.4 | 15.3 | | 8.9 | | 1.3 | |
| **Attending** | | | | | | | |
| Male | 45.9 | 21.4 | 0.377 | 12.5 | 0.006 | 1.4 | 0.371 |
| Female | 54.1 | 23.6 | | 18.2 | | 1.8 | |

All values shown as %, PTSD is shown as scored above cut-off TSQ.
All differences were analysed using $\chi^2$.
HADS, Hospital Anxiety and Depression Scale; PTE, potential traumatic event; PTS, post-traumatic stress; PTSD, post-traumatic stress disorder; TSQ, Trauma Screening Questionnaire.

Statistical Manual of Mental Disorders, 4th edition (DSM-IV) criterion A and E. Of these respondents, 183 reported recognising PTS symptoms after this event (13.4% of the total group, 64.3% of the potentially traumatised group) and 20 screened positive for probable PTSD (1.5% of the total group, 7.0% of the potentially traumatised group, table 2). Of the paediatricians 34.9% reported having experienced a PTE at least 4 weeks ago. This is significantly higher than the percentages of the gynaecologists (12.8%, $\chi^2$, p<0.001) and orthopaedic surgeons (19.5%, $\chi^2$, p<0.001), as seen in table 2. When comparing the prevalence of probable PTSD, the paediatricians had a significant higher prevalence compared with the orthopaedic surgeons (2.2% vs 0.3%, $\chi^2$, p=0.041). Both paediatricians and gynaecologists report to recognise more PTS symptoms.

When comparing gender and position (resident vs attending, table 4), residents are less likely to have experienced a PTE (16.8% vs 22.6%, p=0.030) and thus also report significantly fewer PTS symptoms earlier (7.9% vs 15.6%, p<0.001).

The highest rate of probable PTSD was found among attendings (n=15, 1.6%) compared with three of the residents (0.9%). When comparing gender, male and female physicians report the same number of PTEs, but female physicians are more likely to report earlier PTS symptoms (15.3% vs 11.3%, p=0.044).

Those who witnessed a PTE, reported significantly more defensive work changes after an incident (28.0% vs 20.5%, p=0.007). Respondents who recognised PTS symptoms after an incident considered quitting medical work more often (42.6% vs 29.2%, p=0.029). Those with current probable PTSD significantly considered quitting medical work more often (60.0% vs 35.8%, p=0.032).

When evaluating the presence of a support protocol, this was found not to influence the primary mental health outcomes, except for probable PTSD. Participants who responded that there was no support protocol at their work place after an adverse event, were significantly more likely to report probable PTSD after a PTE (p=0.022).

## DISCUSSION

### Depression and anxiety

This additional analysis combined data from previous work to compare the point prevalence rates of probable depression, anxiety, psychological distress, work-related PTE and PTSD among gynaecologists, orthopaedic surgeons and paediatricians. It is part of the Work-related Adverse and Traumatic Events Research and this paper combines the results of previous studies,[68] to gain statistical power and show differences between support protocols, specialties, gender and position. Our point prevalence rates of depression (6.4%) and anxiety (13.6%) are high compared with 12-month prevalence rates of mood disorders (3.0%) and anxiety (6.0%) in high income populations in the Netherlands, and compared with European 12-month prevalence rate of anxiety (6.4%).[12 13] The point prevalence rates in this study are comparable to American 12-month general prevalence rates of depression (6.7%) and anxiety (18.1%).[29 30] This indicates that physicians are more depressed and anxious compared with the general population and the population with high income in the Netherlands. This high prevalence is in line with current literature, where even higher prevalence rates for depression and anxiety were found among medical students, residents and specialists.[14 15 31] One of the relevant findings of this study is the high prevalence

of psychological distress (17.6%). Scoring above this cut-off is not specific for one diagnosis, but does indicate a high level of distress and a risk for potential unfit physicians. We found a significant increase in the prevalence of depression corresponding with more years in practice. Furthermore, this study showed a significantly higher prevalence of depression and anxiety among female compared with male attendings. This finding correlates with previous study results,[32] where prevalence rates for depression and anxiety were higher among British female doctors compared with males. These findings are also in line with epidemiological data on the gender distribution of mood and anxiety disorders in the general population.[12 13] This finding is relevant, because of the feminisation of medical specialties globally, and an increase in female physicians of 1.0%–2.0% each year in the last 10 years in the Netherlands.[33] These results may implicate that a change is needed in support after adverse events, that is sensitive to the increase in women working in the field of medicine.

### Post-traumatic stress disorder

In this sample, 285 respondents (20.8%) reported to have experienced a PTE that took place longer than 4 weeks ago. This is within the range of the estimated prevalence of second victims of 10.4% and 43.3%, reported by Seys et al.[34] Our prevalence for work-related PTSD (1.5%) is slightly higher, compared with the general Dutch prevalence of PTSD (1.2%).[35] The Dutch prevalence of only work-related PTSD is not known, however, assumed to be lower. Of the potential traumatised respondents, 7.0% screened positive for PTSD, which is comparable to other occupations with mental health hazards, such as police officers after critical police incidents (7.0%).[36]

One of the key findings of this supplementary analysis is that after experiencing a PTE, the absence of a support protocol is significantly associated with more probable PTSD (p=0.022). This once again shows the importance of support protocols and implicates that a well perceived support system may prevent PTS complaints. This survey did, however, not explore the types of support protocols (eg, debriefing vs peer support) in place, as they may differ across hospitals and even within one hospital across specialties. Therefore, nothing can be concluded of the type of support protocol that may help prevent PTS symptoms. However, some form of formalised support may lead to seeking and finding professional help earlier. Studies have shown that the threshold to look for professional mental health as a physician is high. Physicians report a reluctance to seek care, due to barriers such as too little time, fear of losing their medical license and stigma.[37] A (semi) compulsory support protocol after a PTE could potentially increase awareness, lower this threshold and guide physicians in need towards professional support or treatment. Debriefing and hospital based peer support have shown useful and effective after PTEs in several small studies.[10 38] Also, this awareness of more workplace support may lead to perceiving adverse events less

psychotraumatic, however, having a support protocol was not correlated with less PTEs.

Of the residents who filled out the TSQ, three screened positive for PTSD (all female respondents). These findings are in line with the known gender distribution, as women are more likely to develop PTSD than men.[39] Other studies found higher prevalence rates of PTSD (ie, Joseph et al[21] =15%, Mills and Mills[24] =11.9%, Ruitenburg et al[15] =15% and Shi et al[22] =3.8%), but they used different (non-) validated screening instruments and methods. None of these screening instruments assessed when the PTE happened (for a diagnosis of PTSD this needs to be at least 4 weeks ago), meaning that their respondents may not meet criterion A of PTSD and could also be suffering from acute stress symptoms. Furthermore, these studies did not assess whether the potential event was work related, so these numbers show general PTSD prevalence rates which are usually higher.

Studies have shown that dysfunctional coping can predict complicated grief and PTSD severity.[40 41] Thus, it is important to educate physicians about coping mechanisms and stress management techniques to prevent adverse mental health conditions.[42] With social support being an important protective factor for PTSD,[43] the role of educating healthcare professionals to address this problem and support their colleagues in peer-support groups is crucial.[10]

### Job-related consequences of depression, anxiety, adverse events and PTSD

This study shows that physicians consider quitting their job more often, after having experienced PTS symptoms in the past or currently suffering from probable PTSD. Experiencing a PTE is also associated with more defensive medical practice. When looking at the prevalence of complaints at a disciplinary board, the orthopaedic surgeons reported the most complaints (table 1). This may be due to the higher amount of elective work they do, which usually leads to more dissatisfied patients, compared with emergency work. More complaints were not associated with considering quitting work or increased rates of depression, anxiety or psychological distress.

### Strengths

Little research has been done on PTSD in physicians, in particular with regard to mental health in relation to social and occupational functioning and clinical practice-related consequences. We incorporated the A and E criterion in our probable PTSD diagnosis, therefore, our prevalence is more likely to reflect the real prevalence of PTSD. Also, this study included a large number (n=1374) of physicians, which made it possible to analyse gender, position and age differences. The overall response rate among gynaecologists was high (42.9%), followed by the orthopaedic surgeons (29%) and the paediatricians (18.9%).

## Limitations

There are several limitations to this study. The population of this study is limited to three specialties and one country in Europe and thus it is difficult to translate these results to physicians worldwide. However, it is known that working conditions and related stress levels are comparable with the rest of Europe and North America. Also, prevalence rates reported in this study are possibly not comparable to other studies on mental health, as many different samples and measurements are used in mental health studies. Furthermore, avoidance is one of the key symptoms of PTSD and can therefore lead to non-response and non-completion of the survey, as it may trigger symptoms of PTSD. Therefore, the prevalence rates found in psychotrauma studies are often an underestimation of the PTSD prevalence. On the other hand, this self-selection bias may also lead to over-representation of physicians with strong opinions. One suggestion to decrease this non-completion bias is to give respondents the opportunity to finish the survey at a later time point and receive a reminder. The overall response rate was 27.7%, which is slightly higher than the average response rate for email surveys (25.0%)[44] and comparable to similar surveys in this field.[45] Due to the low numbers of PTSD, it was not possible to correlate the type of adverse event with more PTSD. Furthermore, the HADS and TSQ are validated screening instruments, however, they only give an estimate of the prevalence rates of probable disorders as they do not cover all the symptom domains of the disorders. Only the prevalence of PTSD is work related, the numbers for depression and anxiety are general prevalence rates.

## CONCLUSION

Physicians are often exposed to PTEs over the course of their career with potentially high emotional impact. The prevalence rates for probable depression (6.4%) and anxiety (13.6%) were higher among physicians, compared with the general population with high income in the Netherlands. The point prevalence of clinical psychological distress was 19.0% and of probable PTSD was 1.5%. Female physicians are significantly more prone to develop depressive and anxiety disorders than their male colleagues. PTEs were associated with more defensive practice and dissatisfaction about work. And importantly, the absence of a support protocol was associated with more probable PTSD. This once again shows the importance of support protocols and implicates that a well-perceived support system, such as peer support and debriefings, may prevent PTS complaints and unfit physicians.

## Author affiliations
[1]Psychiatry, Amsterdam UMC - Locatie AMC, Amsterdam, The Netherlands
[2]Faculty of Medicine, Amsterdam UMC - Locatie AMC, Amsterdam, The Netherlands
[3]Obstetrics and Gynaecology, OLVG Locatie Oost, Amsterdam, The Netherlands
[4]Obstetrics and Gynaecology, Amsterdam UMC - Locatie AMC, Amsterdam, The Netherlands
[5]Psychotrauma Research, Arq National Psychotrauma Center, Diemen, North Holland, The Netherlands

**Acknowledgements** We would like to thank the Dutch Society of Obstetrics & Gynaecology (NVOG), Dutch Society of Orthopaedic Surgeons (NOV), the Dutch Society of Paediatricians (NvK), and their members, for their cooperation in this study.

**Contributors** KWFS, MEvS, CAIS and MGvP were involved in the design of this study. KWFS and HSP analysed the data and made the first draft. KWFS, HSP, MEvS, CAIS, MO and MGvP were involved in finalising the manuscript.

**Funding** The authors have not declared a specific grant for this research from any funding agency in the public, commercial or not-for-profit sectors.

**Competing interests** None declared.

**Patient and public involvement** Patients and/or the public were not involved in the design, or conduct, or reporting, or dissemination plans of this research.

**Patient consent for publication** Not required.

**Ethics approval** This study was exempted from ethical approval by the Medical Research Ethics Committees United (MEC-U), and registered under numbers W17.169 and W18.096.

**Provenance and peer review** Not commissioned; externally peer reviewed.

**Data availability statement** Data are available on reasonable request. The data can be made available by the corresponding author on reasonable request and for a well-defined purpose. The request will be discussed in our research group. The data will be de-identified participant data. A statistical analysis plan is available.

**ORCID iDs**
Karel Willem Frank Scheepstra http://orcid.org/0000-0003-4970-057X
Maria G van Pampus http://orcid.org/0000-0002-3020-8908

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
