## [Reviewer comments · BMJ Open]

ARTICLE DETAILS

TITLE (PROVISIONAL)	Potential traumatic events in the workplace and depression, anxiety and posttraumatic stress: a cross-sectional study among Dutch gynaecologists, paediatricians and orthopaedic surgeons.
AUTHORS	Scheepstra, Karel; Pauw, Hannah; van Steijn, Minouk; Stramrood, Claire; Olf, Miranda; van Pampus, Maria G.

VERSION 1 – REVIEW

REVIEWER	Muhammad Naseem Khan Khyber Medical University, Pakistan
REVIEW RETURNED	21-Sep-2019

GENERAL COMMENTS	Page. 7, Line 47-49: The orthopaedic surgeons reported to have had the most complaints at a disciplinary board. This is not obvious from the data as you explained in the lines following. No supplementary file 3 is provided although cited in the manuscript? Page 10, line 26-30: When comparing gender, male and female physicians report the same number of PTEs, but female physicians are more likely to report earlier PTS-symptoms (15.3% vs. 11.3%, $p=0.044$). These figures are not reflected in the table?
---

REVIEWER	N. Zoe Hilton University of Toronto and Waypoint Research Institute
REVIEW RETURNED	22-Nov-2019

GENERAL COMMENTS	This manuscript describes analysis of data from gynecologists, pediatricians, and orthopedic surgeons on their exposure to adverse events and potentially traumatic events in the workplace, as well as scores on screeners for depression, anxiety, psychological distress (a combination of depression and anxiety scores), and posttraumatic stress. Comparisons are made across groups of male vs female, resident vs attending, and work experience (supplementary tables). I find the study interesting and the methods are generally well presented. My major concerns have to do with the nature of the study as a re-analysis of work already published, the reliance on descriptive and cross-tab analyses, and the need for more discussion of limitations and implications. 1. In the discussion section on page 10 we are told that the present study is part of a larger project and combines data from previously
--

	published studies to look at certain variables: protocol, specialty, gender, position. a. This information should be provided much earlier. In the abstract, state that this study is a supplementary analysis of previous work. b. In the introduction, review the previous published work. c. In the introduction, and especially in the paragraph before the methods section, explain how this analysis differs from previously published works with these data. Provide a rationale for the need to look again at the data for these variables. Include a clear research question and objectives, in order to justify and contextualize the study. d. In the discussion, it is stated that the present study aimed to look at protocol, specialty, gender, position. Gender and specialty differences in particular are presented/calculable from the previous works. I don't see results examining the association between outcomes and support protocols in this manuscript, except for the percentages of different specialties reporting that they have a support protocol. Therefore, a stronger rationale is required. e. In the discussion, acknowledge the overlap with previous publications and address the implications for the current study's contribution to the literature. 2. Results are tabulated as percentages and statistically significant results are reported in terms of p values. The data could be used with greater effect and the presentation of existing results could be improved. a. Present actual chi squared values in the tables, not just p values. b. Show chi squared values for both non-significant and significant results. c. Table titles could be more informative; specifically, state the nature of the data that are being presented (my interpretation is that data are percent scoring above the cut-off on the screening tools, but this should be evident from the title and/or column headings). d. Were any adjustments used for the number of tests conducted? If yes, describe the adjustment, if not, please state so in the text. e. Consider the potential for more sophisticated research questions and analyses. For example, how are participant characteristics, adverse events, and traumatic events, correlated with mental health measures, and which variables are independent, unique contributors in a regression model of mental health outcomes? f. Support is not really explored, even though the discussion focuses on the importance of supports. Is having workplace support related to the nature/prevalence of adverse events and traumatic events? Or to the mental health outcomes? 3. In the discussion, additional limitations could be acknowledged and suggestions for overcoming limitations should be offered. a. It is good that criteria A and F were considered, but I note that the
--	---

	PTSD screening tool does not cover all the PTSD symptom domains well, mostly re-experiencing and arousal. This should be noted as a limitation. b. On page 13, limitations, it is stated that “avoidance is one of the key symptoms of PTSD and can therefore lead to non-response and underestimation.” This sentence is not clear and needs explaining. Do you mean to point out that avoidance is not measured adequately on the PTSD screening tool used, so PTSD symptoms may be underestimated? Or perhaps that potential participants might avoid the study because of its subject matter? What are some research solutions to this limitation that could be implemented in future studies? c. It should be noted that comparisons with other surveys, especially in the prevalence of disorders compared with the Netherlands population, could be limited by differences in sampling, measurement, etc. d. A few times it is mentioned that support is needed. Can you describe the kind of support that is needed? This could be connected with the literature on debriefing, workplace wellness programs, social support, clinical assessment and treatment, or other kinds of support. e. On page 11, end of first paragraph, there is a sentence about the increase of women in medicine that is followed by a sentence about the potential for unfit physicians. Unfortunately a reader might infer that you are connecting female physicians with unfit physicians. This needs some clarification and qualification. This is an opportunity to discuss the implications for support that is sensitive to the increase in women and other non-traditional groups working in the field of medicine. 4. The survey gathered data on various workplaces including ER, NICU, etc. Did work setting have any relation to workplace events or mental health? 5. In the Methods section on measurements, please say in what way the HADS and TSQ are validated. Have they demonstrated concurrent validity with larger assessment tools or clinical diagnosis?
--	--

VERSION 1 – AUTHOR RESPONSE

Reviewer: 1

Reviewer Name: Muhammad Naseem Khan

Institution and Country: Khyber Medical University, Pakistan

Please state any competing interests or state 'None declared': None declared

Please leave your comments for the authors below

Page. 7, Line 47-49: The orthopaedic surgeons reported to have had the most complaints at a disciplinary board.

This is not obvious from the data as you explained in the lines following.

- The number of complaints at a disciplinary board can be seen in table 1. We agree that this may not be a clear sentence and added the prevalence of complaints in the sentence, as following: 'The orthopaedic surgeons reported to have had the most complaints at a disciplinary board (29.8%).' We also noticed that in table 1 these results of complaints were presented somewhat unclear, we changed this.

No supplementary file 3 is provided although cited in the manuscript?

- Thank you for your observation, this should have been supplementary file 2; we changed it accordingly.

Page 10, line 26-30: When comparing gender, male and female physicians report the same number of PTEs, but female physicians are more likely to report earlier PTS-symptoms (15.3% vs. 11.3%, $p=0.044$).

These figures are not reflected in the table?

- The numbers mentioned (15.3% female vs. 11.3% male) are shown in table 4. We agree that the table was not clear enough, we therefore changed the layout of tables 3 and 4 entirely.

Reviewer: 2

Reviewer Name: N. Zoe Hilton

Institution and Country: University of Toronto and Waypoint Research Institute

Please state any competing interests or state 'None declared': None declared other than shared research interests

Please leave your comments for the authors below

This manuscript describes analysis of data from gynecologists, pediatricians, and orthopedic surgeons on their exposure to adverse events and potentially traumatic events in the workplace, as well as scores on screeners for depression, anxiety, psychological distress (a combination of depression and anxiety scores), and posttraumatic stress. Comparisons are made across groups of male vs female, resident vs attending, and work experience (supplementary tables). I find the study interesting and the methods are generally well presented. My major concerns have to do with the nature of the study as a re-analysis of work already published, the reliance on descriptive and cross-tab analyses, and the need for more discussion of limitations and implications.

1. In the discussion section on page 10 we are told that the present study is part of a larger project and combines data from previously published studies to look at certain variables: protocol, specialty, gender, position.

a. This information should be provided much earlier. In the abstract, state that this study is a supplementary analysis of previous work.

- We agree that readers may not immediately recognize the rationale for an analysis of our combined data. It may seem like a re-analysis of previous work, however the goal of this combined study is to compare across groups (different specialty, gender, position, etc) and look at clinical practice related consequences with a large number of respondents. Combining the data of previous work allows us to explore the association between support protocols and mental health symptoms in a larger group, and gain statistical power. We acknowledge your

major concern with the nature of this study, we therefore put meticulous effort in addressing this point.

We have now added in the abstract: "Objective: To compare the prevalence of work-related potential traumatic events, support protocols and mental health symptoms across Dutch gynaecologists, orthopaedic surgeons and paediatricians." "Design: Cross-sectional study, supplementary analysis of combined data."

We added in the methods section: "This is an additional analysis of combining data of previous work."

b. In the introduction, review the previous published work.

- We have added some of the most important findings of previous published work in the introduction. We added: "However, our previous work showed that most gynaecologists, paediatricians and midwives report to experience support and protocols after work-related potential traumatic events as insufficient. Dutch paediatricians report 'missing a diagnosis' as having the most emotional impact and 'aggressive behaviour from parents' as the most common adverse event. [6] Midwives working in primary care reported higher levels of anxiety compared with their colleagues working in secondary or tertiary care.[7] Gynaecologists report that development of coping strategies with work-related stressors are not educated during- or part of medical and specialist training.[8]"

c. In the introduction, and especially in the paragraph before the methods section, explain how this analysis differs from previously published works with these data. Provide a rationale for the need to look again at the data for these variables. Include a clear research question and objectives, in order to justify and contextualize the study.

- We agree that, from the objective stated in the original manuscript, it was not entirely clear what the research questions add on top of our previous work. We changed it according to your suggestions, emphasizing the design is a supplementary analysis with additional research questions as mentioned earlier.
Therefore we changed the end of the introduction into: "The main objective of this supplementary analysis is to compare the prevalence of potential traumatic events (PTEs) and related mental health between a large group of gynaecologists, paediatricians and orthopaedic surgeons and show differences between gender, age and position. Additionally, this study investigates mental health in relation to social and occupational functioning and clinical practice related consequences. Combining the data of previous work allows us to gain statistical power to explore the association between support protocols and mental health symptoms in a larger group."

d. In the discussion, it is stated that the present study aimed to look at protocol, specialty, gender, position. Gender and specialty differences in particular are presented/calculable from the previous works. I don't see results examining the association between outcomes and support protocols in this manuscript, except for the percentages of different specialties reporting that they have a support protocol. Therefore, a stronger rationale is required.

- We agree that some of the results may be calculable from previous work, however the data from orthopaedic surgeons had not been published yet. Also, combining the data has given additional statistical power to show significant associations between variables and the small group of probable PTSD. This was not possible in previous work due to low numbers of PTSD. This rationale is added in the methods section. You are right that the association

between lacking a protocol and significant more PTSD is not reported in the results or in the learning points. It is however presented in the abstract and discussion as a key finding. As it is a direct result of combining the data and a key finding, we added the association between outcomes and support protocols in the results.

In the results section it now reads: " When evaluating the presence of a support protocol, this was found not to influence the primary mental health outcomes, except for probable PTSD.

Participants who responded that there was no support protocol at their work-place after an adverse event, were significantly more likely to report probable PTSD after a PTE ($p=0.022$)."

e. In the discussion, acknowledge the overlap with previous publications and address the implications for the current study's contribution to the literature.

- In the discussion, we acknowledged the overlap with previous work and laid more emphasis on what this analysis contributes to literature. Now the start of the discussion reads: "This additional analysis combined data from previous work to compare the point prevalence rates of probable depression, anxiety, psychological distress, work-related PTE and PTSD among gynaecologists, orthopaedic surgeons and paediatricians. It is part of the Work-related Adverse and Traumatic Events Research (WATER) and this paper combines the results of several previous studies [6,8], to gain statistical power and show differences between support protocols, specialties, gender and position."

2. Results are tabulated as percentages and statistically significant results are reported in terms of p values. The data could be used with greater effect and the presentation of existing results could be improved.

a. Present actual chi squared values in the tables, not just p values.

b. Show chi squared values for both non-significant and significant results.

- We changed tables 3 and 4 entirely to make sure the presentation of the results is improved. Also we added chi squared values in the tables for all results.

c. Table titles could be more informative; specifically, state the nature of the data that are being presented (my interpretation is that data are percent scoring above the cut-off on the screening tools, but this should be evident from the title and/or column headings).

- We rephrased the titles of the tables and added that results are displayed as % scored above the validated cut off values. Cut off values are described in the method section and these are not repeated in the tables, to make sure the tables do not contain too much information and remain easily interpretable.

The table titles now read: "Table 2: Depression, anxiety, psychological distress and PTSD by specialty." "Table 3: Depression, anxiety and psychological distress by position (resident vs. attending) and gender." "Table 4: Posttraumatic stress outcomes by position (resident vs. attending) and gender."

d. Were any adjustments used for the number of tests conducted? If yes, describe the adjustment, if not, please state so in the text.

- Multiple testing corrections were not made and we have added this to our methods section.

e. Consider the potential for more sophisticated research questions and analyses. For example, how are participant characteristics, adverse events, and traumatic events, correlated with mental health measures, and which variables are independent, unique contributors in a regression model of mental health outcomes?

- We did perform more sophisticated analyses with a logistic regression on several variables looking for associations with mental health outcomes. As described in the results, we mainly found working experience as a contributor to both anxiety and depression, but not for other variables. We added to the methods – statistics section: “A multivariable logistic regression analysis was performed to estimate the effects of age, work-experience, gender and position on the likelihood that participants develop mental health outcomes, a p-value equal to or less than 0.05 was considered statistically significant.”

f. Support is not really explored, even though the discussion focuses on the importance of supports. Is having workplace support related to the nature/prevalence of adverse events and traumatic events? Or to the mental health outcomes?

- This manuscript did not focus on type of support protocol in place at various hospitals, only on whether there is a formalized support protocol after an adverse event. Until now, we had only looked into the relation between a protocol and mental health outcomes, however your suggestion that it may also work the other way around is interesting; having more support in the workplace may lead to perceiving adverse events as less traumatic and therefore reduce the prevalence of PTEs. We looked into this but could not verify this hypothesis with a statistical difference. We added more on support, see point 3d. We added in the discussion – PTSD section: “Also, this awareness of more workplace support may lead to perceiving adverse events less psychotraumatic, however having a support protocol was not correlated with less PTEs.”

3. In the discussion, additional limitations could be acknowledged and suggestions for overcoming limitations should be offered.

a. It is good that criteria A and F were considered, but I note that the PTSD screening tool does not cover all the PTSD symptom domains well, mostly re-experiencing and arousal. This should be noted as a limitation.

- The TSQ is a validated screening instrument for PTSD, however you are right that not all symptom domains are in the TSQ. This is the same, however less obvious, for the HADS. We added this to the limitations section, it now reads: “Furthermore, the HADS and TSQ are validated screening instruments, however they only give an estimate of the prevalence rates of probable disorders as they do not cover all the symptom domains of the disorders.”

b. On page 13, limitations, it is stated that “avoidance is one of the key symptoms of PTSD and can therefore lead to non-response and underestimation.” This sentence is not clear and needs explaining. Do you mean to point out that avoidance is not measured adequately on the PTSD screening tool used, so PTSD symptoms may be underestimated? Or perhaps that potential participants might avoid the study because of its subject matter? What are some research solutions to this limitation that could be implemented in future studies?

- What we meant was that filling out a survey with questions about a potential traumatic event may trigger posttraumatic stress symptoms. Therefore this may lead to non completion of the survey of potential traumatized respondents, and thus an underestimation of the prevalence. We agree that this may not have been clear, therefore we made this part more comprehensive and added some suggestions to overcome this bias, it now reads:

"Furthermore, avoidance is one of the key symptoms of PTSD and can therefore lead to non-response and non-completion of the survey, as it may trigger symptoms of PTSD. Therefore the prevalence rates found in psychotrauma studies are often an and underestimation of the PTSD-prevalence. On the other hand, this self-selection bias may also lead to overrepresentation of physicians with strong opinions. One suggestion to decrease this non-completion bias, is to give respondents the opportunity to finish the survey at a later time point and receive a reminder."

c. It should be noted that comparisons with other surveys, especially in the prevalence of disorders compared with the Netherlands population, could be limited by differences in sampling, measurement, etc.

- We added this to our limitations as following: "Also, prevalences reported in this study are possibly not comparable to other studies on mental health, as many different samples and measurements are used in mental health studies."

d. A few times it is mentioned that support is needed. Can you describe the kind of support that is needed? This could be connected with the literature on debriefing, workplace wellness programs, social support, clinical assessment and treatment, or other kinds of support.

- Most evidence exists for debriefing and hospital based peer-support. We added a large part on support and the function of a protocol in the discussion, and a few words in the conclusion. We added to the discussion: "This survey did however not explore the types of support protocols (e.g. debriefing versus peer-support) in place, as they may differ across hospitals and even within one hospital across specialties. Therefore nothing can be concluded of the type of support protocol that may help prevent PTS symptoms. However, some form of formalised support may lead to seeking- and finding professional help earlier. Studies have shown that the threshold to look for professional mental health as a physician is high. Physicians report a reluctance to seek care, due to barriers such as too little time, fear of losing their medical license and stigma.[37] A (semi) compulsory support protocol after a PTE could potentially increase awareness, lower this threshold and guide physicians in need towards professional support or treatment. Debriefing and hospital based peer-support have shown useful and effective after PTEs in several small studies. [10,38]" And to the conclusion: "This once again shows the importance of support protocols and implicates that a well perceived support system, such as peer support and debriefings, may prevent posttraumatic stress complaints and unfit physicians."

e. On page 11, end of first paragraph, there is a sentence about the increase of women in medicine that is followed by a sentence about the potential for unfit physicians. Unfortunately a reader might infer that you are connecting female physicians with unfit physicians. This needs some clarification and qualification. This is an opportunity to discuss the implications for support that is sensitive to the increase in women and other non-traditional groups working in the field of medicine.

- We understand your worries about how readers might interpret these results. It was not the intention to qualify female physicians as unfit and we want to avoid such an interpretation or implication in any way. We have considered your suggestion for further clarification, moved the sentence on unfit physicians and changed these sentences accordingly. It now reads: "This finding is relevant, because of the feminization of medical specialties globally, and an increase in female physicians of 1.0 to 2.0 % each year in the last ten years in the Netherlands.[33] These results may implicate that a change is needed in support after adverse events, that is sensitive to the increase in women working in the field of medicine."

4. The survey gathered data on various workplaces including ER, NICU, etc. Did work setting have any relation to workplace events or mental health?

- This survey indeed gathered some information on work setting of paediatricians, however not for the ObGyn and surgeons. We did not find any relation between work setting and mental health among paediatricians. As the main objective of this manuscript was to compare across specialties, this was not added to the manuscript, it is however interesting for further research.

5. In the Methods section on measurements, please say in what way the HADS and TSQ are validated. Have they demonstrated concurrent validity with larger assessment tools or clinical diagnosis?

- In the methods – measurements section the psychometric properties of the HADS and TSQ are described in terms of specificity, sensitivity and cronbach alpha. Both the HADS and TSQ have been validated in Dutch populations and were compared with the golden standard of the clinical diagnosis (e.g. a CAPS interview for diagnosing PTSD), as can be read in the referenced papers.

VERSION 2 – REVIEW

REVIEWER	Muhammad Naseem Khan Khyber Medical University, Peshawar, Pakistan
REVIEW RETURNED	22-Feb-2020

GENERAL COMMENTS	I have reviewed this paper before in September 2019, however I couldn't find any point by point changes accompanying this revision. The overall quality of the paper is quite poor specifically the way the results are written and described. The topic area is very important and there are areas which can be improved. These are specifically  1. Some of the results are described in the paper, however there is no file etc. showing these results. 2. Again some of the tables are described however there isn't any such results present in the table. 3. I would also like to see more details regarding some of the tools like the way PTEs, and PTS were measured and scored.
--

REVIEWER	Zoe Hilton Waypoint Centre for Mental Health Care Canada
REVIEW RETURNED	21-Feb-2020

GENERAL COMMENTS	I have reviewed this revised manuscript in light of my comments on the original version. Although I was unable to locate the point by point letter of response to reviewer comments, I find that the authors have been responsive to the comments by providing additional information to the text and tables.  1. The authors are now give clearer details about their regression analysis. On page 10, about half way through paragraph two there are results reported that appear to come from the regression
---

	analysis. It would be helpful to flag this for the reader, perhaps by a phrase such as, "Turning to the results of the regression analysis...". I would also like to see the overall regression model statistics in the main text.
--	--

VERSION 2 – AUTHOR RESPONSE

Reviewer Name: Zoe Hilton

Institution and Country:

Waypoint Centre for Mental Health Care

Canada

Please state any competing interests or state 'None declared': None stated

Please leave your comments for the authors below I have reviewed this revised manuscript in light of my comments on the original version. Although I was unable to locate the point by point letter of response to reviewer comments, I find that the authors have been responsive to the comments by providing additional information to the text and tables.

1. The authors are now give clearer details about their regression analysis. On page 10, about half way through paragraph two there are results reported that appear to come from the regression analysis. It would be helpful to flag this for the reader, perhaps by a phrase such as, "Turning to the results of the regression analysis...". I would also like to see the overall regression model statistics in the main text.

- Thank you for your comment. We introduced our regression statistics as suggested and expanded the model statistics. It now reads: 'Turning to the results of the logistic regression analysis, there was a significant increase in the prevalence of depression in the groups with more years in practice (OR 2.95, p=0.030, 95% CI 0.107 - 2.054, supplementary file 2), but not with increased age. We found a significant decrease in the prevalence of anxiety with more years in practice (OR 0.33, p=0.001, 95% CI -1.784 - 0.448, supplementary file 2).'

Reviewer: 1 *additional comments included in the attached file*

Reviewer Name: Muhammad Naseem Khan

Institution and Country: Khyber Medical University, Peshawar, Pakistan Please state any competing interests or state 'None declared': None declared

Please leave your comments for the authors below I have reviewed this paper before in September 2019, however I couldn't found any point by point changes accompanying this revision. The overall quality of the paper is quite poor specifically the way the results are written and described. The topic area is very important and there are areas which can be improved.

- Thank you for your valuable comments. We have meticulously tried to improve the quality of the paper and changed the text according to your suggestions. We also added our response to your in-text comments below.

These are specifically

1. Some of the results are described in the paper, however there is no file etc. showing these results.

- We agree that for some of the results it was unclear in which table these could be found. Therefore we tried to clarify this and refer to the correct table. For example in the results it now reads: 'In all specialties most of the respondents were attending physicians, as seen in table 1.'

2. Again some of the tables are described however there isn't any such results present in the table.

- Thank you for pointing out that it may be unclear where some of the results can be found. Nearly all results described in the main text that can be found in a table, for all these results we tried to refer to the table in the main text. For example it now reads: 'This is significantly higher than the percentages of the gynaecologists (12.8%, $p < 0.001$) and orthopaedic surgeons (19.5%, $p < 0.001$), as seen in table 2.' Only for a few outcomes on clinical consequences (defensive work changes and considering quitting work) the results are only described in the main text.

3. I would also like to see more details regarding some of the tools like the way PTEs, and PTS were measured and scored.

- We agree that it may have been unclear how the number of PTEs and previous PTS-symptoms were quantified. For both the PTEs and symptoms in the past questions were used based on the criteria of PTSD according to the DSM-IV. To clarify these measurements, we elaborated on this in the methods section by adding: 'To quantify the number of PTEs and previous PTS-symptoms, questions were added based on the criteria of the diagnosis of PTSD in the DSM-IV (questions 22 and 35, supplementary file 1).'

In-text comments reviewer 1:

Abstract

1. The methods should be in the past tense. (page 2)

- We have changed this according to your track changes.

2. The conclusion in this sentence 'Adverse events were associated with more dissatisfaction with work and a more defensive practice' is not mentioned in your results in the abstract above? (page 2)

- It is correct that the interpretation 'dissatisfaction with work' is not mentioned in the results section, an increase in defensive practice and considering quitting medical work are mentioned in the results. We therefore rephrased this and it now reads: 'Adverse events were associated with considering to quit medical practice and a more defensive practice.'

Introduction

3. Not clear what you want here: 'However, our previous work showed that most gynaecologists, paediatricians and midwives report to experience support and protocols after work-related potential traumatic events as insufficient.' Please rewrite for clarity. (page 4)

- Reviewer 1 suggested adding results from our previous work, to give additional context to this manuscript. We agree that this sentence may not have been clear, we therefore added that adequate coping does not warrant successfully dealing with very severe medical incidents. This part now reads: 'However, adequate coping mechanisms do not necessarily warrant successfully coping with severe potential traumatic events. Our previous work showed that most gynaecologists, paediatricians and midwives report to experience support and protocols after work-related potential traumatic events as insufficient. Dutch paediatricians reported 'missing a diagnosis' as having the most emotional impact and 'aggressive behaviour from parents' as the most common adverse event. [6] Midwives working in primary care reported higher levels of anxiety compared with their colleagues working in secondary or tertiary care.[7] Gynaecologists reported that development of coping strategies with work-related stressors are neither not educated taught during graduate nor in during- or part of medical and

specialist training.[8]'

4. The results in this sentence: 'Additionally, this study investigates mental health in relation to social and occupational functioning and clinical practice related consequences.' Are not shown in results section. (page 5)

- You are correct that part of these outcomes are not described in the results section. We mainly described clinical practice related consequences in relation to traumatic events and we therefore changed this sentence into: 'Additionally, this study investigates PTEs and clinical practice related consequences.'

Methods

5. Can you add information regarding how HADS was scored? (page 7)

- We added that the HADS uses a 4-point Likert scale and that we used the cut off scores according to these Dutch validation studies. It now reads: 'The HADS is a 14-item screening instrument for depression and anxiety with 14 items (4-likert scale), where the subscales contain seven questions each. The cut-off value of the Dutch version of the depression (HADS-D) and anxiety subscale (HADS-A) is equal to or higher than eight. The total HADS cut-off value for general distress is equal or higher than 12, corresponding to clinically relevant psychological distress; this is a rather non-specific but sensitive measurement to screen for mental disorders.[26] Cut off scores were used according to these validation studies.'

•

6. Higher cut offs (TSQ) will change the sensitivity and specificity? (page 7)

- It may have been unclear that a cut-off score of six was meant. We therefore changed this sentence and it now reads: 'The TSQ is a ten-item screening instrument for a probable diagnosis of PTSD according to the DSM-IV, with a sensitivity of 0.94 and specificity of 0.56 when using a cut-off of equal to six or higher.'

7. Only three specialities? (page 7)

- This was changed according to your suggestion, it now reads: 'These qualitative questions were piloted among several medical specialists of all three medical specialties, to increase face value of the questions.'

8. Write JASP in Full the first time. (page 7)

- Thank you for this observation. JASP is not an abbreviation and is written in full as such.

Results section

9. What do you mean by majority? As you called 85% majority as well as 60% or more? Be specific. (page 8)

- We understand that the term majority may be unclear, we therefore changed this sentence, added numbers and made a referral to the table. It now reads: 'In all specialties most (62.3 – 74.9%) of the respondents were attending physicians, as seen in table 1.'

10. Referral to the table is incorrect: Not only HADS, the table has PTSD as well (page 8).

- It was added that the outcomes of the TSQ are also shown in table 2: 'Outcomes of the HADS and TSQ by specialty are shown in table 2.'

11. Not needed to report chi-square values in the table (table 3, page 9).

- Reviewer 2 suggested adding chi square values, therefore we added statistics in revision 1.

12. Which test applied as not shown in the table, in sentence 'However, looking at the prevalence of anxiety and psychological distress, the orthopaedic surgeons scored significantly lower compared to both gynaecologists and paediatricians.' (page 10)

- We agree that the statistics of these results were somewhat unclear, we therefore added numbers and statistics. Chi-square was used, as described in the methods section. It now reads: 'However, looking at the prevalence of anxiety, the orthopaedic surgeons scored significantly lower compared to both gynaecologists (15.8% vs. 8.2%, $p < 0.01$) and paediatricians (14.1% vs. 8.2%, $p < 0.05$). Also psychological distress was significantly lower compared to gynaecologists (18.2% vs. 12.0%, $p < 0.05$) and paediatricians (19.3% vs. 12.0%, $p < 0.01$).'

13. Not presented in the main results? (regression analysis, page 10)

- These results are shown in the supplementary files, otherwise the number of tables is too high. Also the statistics were added for more clarification. It now reads: 'Turning to the results of the logistic regression analysis, there was a significant increase in the prevalence of depression in the groups with more years in practice (OR 2.95, $p = 0.030$, 95% CI 0.107 - 2.054, supplementary file 2), but not with increased age. We found a significant decrease in the prevalence of anxiety with more years in practice (OR 0.33, $p = 0.001$, 95% CI -1.784 - 0.448, supplementary file 2).'

14. Not present in results? (concerning prevalence PTEs, page 11)

- These prevalences can be found in table 2, therefore we added a reference: 'This is significantly higher than the percentages of the gynaecologists (12.8%, $p < 0.001$) and orthopaedic surgeons (19.5%, $p < 0.001$), as seen in table 2.'

15. Not present in the main results (concerning sentence on clinical consequences of PTEs, page 13).

- Thank you for pointing out that it may be unclear where some of the results can be found. Nearly all results described in the main text that can be found in a table, for all these results we tried to refer to the table in the main text. Only for a few outcomes on clinical consequences (defensive work changes and considering quitting work) the results are only described in the main text.

16. Poorly written, can't conclude (concerning sentence on clinical consequences of PTEs, page 13).

- We agree that this paragraph was difficult to read, we therefore rephrased three sentences. It now reads: 'Those who witnessed a PTE, reported significantly more defensive work changes after an incident (28.0% vs 20.5%, $p = 0.007$). Respondents that recognised PTS-symptoms after an incident considered quitting medical work more often (42.6% vs. 29.2%, $p = 0.029$). Those with current probable PTSD significantly considered quitting medical work more often (60.0% vs. 35.8%, $p = 0.032$).'

VERSION 3 – REVIEW

REVIEWER	Muhamad Naseem Khan Khyber Medical University, Pakistan
REVIEW RETURNED	12-Apr-2020

GENERAL COMMENTS	There are still areas which needs improvement and some of the comments I raised are still not responded in either way. Please respond to these queries and solve these concerns.
--

VERSION 3 – AUTHOR RESPONSE

Reviewer: 1

Reviewer Name: Muhamad Naseem Khan

Institution and Country: Khyber Medical University, Pakistan

Please state any competing interests or state 'None declared': None

Please leave your comments for the authors below

There are still areas which needs improvement and some of the comments I raised are still not responded in either way. Please respond to these queries and solve these concerns.

Comment 1: Not mentioned in your results in the abstract above? Still not addressed.

- The part on defensive practice and considering giving up medical work was added to the result part of the abstract. We added to the abstract: 'The absence of a support protocol was significantly associated with more probable posttraumatic stress disorder ($p=0.022$). Those who witnessed a PTE, reported more defensive work changes (28.0% vs 20.5%, $p=0.007$) and those with probable posttraumatic stress disorder considered to quit medical work more often (60.0% vs. 35.8%, $p=0.032$).'

Comment 2: Not clear what you want here. Please rewrite for clarity. I suggested to rewrite for better understanding, but still the same?

- This part was added based on the comments of reviewer Zoe Hilton, to describe how this study is an addition on our previous work. We strongly feel this introduction is needed to create context for this manuscript, to show this work is not just a redo of earlier work. However we tried to shorten it somewhat to make sure it is more readable. It now reads: 'However, adequate coping mechanisms do not warrant successful coping with severe potential traumatic events. Our previous work showed that most gynaecologists, paediatricians and midwives experience support and protocols after work-related potential traumatic events as insufficient. [6,7]. Dutch gynaecologists reported that development of coping strategies with work-related stressors are neither taught during graduate, nor in specialist training.[8]'

Comment 3: Not needed to report chi-square values in the table. I am still not convinced as most of the journals don't report test values.

- We added these values based on the comments of reviewer Zoe Hilton, however we agree most journals do not report these values. We therefore removed the chi-square values and added that chi-square test was performed below the table. The same changes were made for table 4.

Comment 4: Which test applied as not shown in the table?

- We agree that the statistics of these results were somewhat unclear, we therefore added numbers and statistics. Chi-square was used, and we added this to the sentence. We did not add p-values for the whole table, since the table would contain an excessive amount of information. It now reads: 'However, looking at the prevalence of anxiety, the orthopaedic surgeons scored significantly lower compared to both gynaecologists (15.8% vs. 8.2%, χ^2 , $p<0.01$) and paediatricians (14.1% vs. 8.2%, χ^2 , $p<0.05$). Also psychological distress was significantly lower compared to gynaecologists (18.2% vs. 12.0%, χ^2 , $p<0.05$) and paediatricians (19.3% vs. 12.0%, χ^2 , $p<0.01$).'

Comment 5 and 6: This confidence interval is not statistically significant as has a value of 1 in it and even this has'nt been reported in the supplementary file.

- You are right that the CI was not shown in the supplementary file. The numbers shown in this log

regression are lower- and upper bound of a confidence interval of 95%. We changed this to CI OR ratio scale to make it easier to interpret and added the values in the supplementary file. It now reads: 'Turning to the results of the logistic regression analysis, there was a significant increase in the prevalence of depression in the groups with more years in practice (OR 2.95, p=0.030, 95% CI 1.11 - 7.80, supplementary file 2), but not with increased age. We found a significant decrease in the prevalence of anxiety with more years in practice (OR 0.33, p=0.001, 95% CI 0.17 - 0.64, supplementary file 2).'

Comment 7: Not present in results? The p values are not there.

• You are correct that the numbers are in the table, but not p-values. When adding p-values to this table, this table will have too much information and becomes unreadable. We therefore only describe significant results of this table in the results paragraph. We however added that chi-square tests were performed, to clarify which tests were applied. It now reads: 'This is significantly higher than the percentages of the gynaecologists (12.8%, χ^2 , p<0.001) and orthopaedic surgeons (19.5%, χ^2 , p<0.001), as seen in table 2. When comparing the prevalence of probable PTSD, the paediatricians had a significant higher prevalence compared to the orthopaedic surgeons (2.2% vs. 0.3%, χ^2 , p=0.041).'

VERSION 4 – REVIEW

REVIEWER	Muhammad Naseem Khan Institute of Public Health & Social Sciences, Khyber Medical University, Peshawar
REVIEW RETURNED	29-May-2020
GENERAL COMMENTS	This is my 3rd review of the paper, however still there were queries which were not addressed as highlighted in the 3rd version with my comments in the 2nd version.